# Homes Heat Health protocol: an observational cohort study measuring the effect of summer temperatures on sleep quality

Kevin Lomas ![ORCID], Kevin Morgan ![ORCID], Victoria Haines ![ORCID], Iuliana Hartescu ![ORCID], Arash Beizaee ![ORCID], Jo Barnes ![ORCID], Zoe Zambelli ![ORCID], Monisha Ravikumar ![ORCID], Vincenzo Rossi ![ORCID]

Loughborough University, Loughborough, UK

**Correspondence to**
Professor Kevin Lomas;
k.j.lomas@lboro.ac.uk

## ABSTRACT

**Introduction** Quality sleep is essential to our health and well-being. Summertime temperatures in the bedrooms of homes in temperate climates are increasing, especially in city apartments. There is very little empirical evidence of the effect of temperature on sleep when people are sleeping in their own bedroom. The Homes Heat Health project seeks to develop a measurable definition of temperature-related sleep disturbance and the effects on health, and so produce a credible criterion for identifying overheating in new and existing homes.

**Methods and analysis** A cohort of at least 95 people that live in London apartments and who are free of significant personal and health factors that could affect sleep are being recruited for an ongoing observational cohort study. A baseline questionnaire determines their customary sleep patterns and health. The geometrical form and thermal characteristics of their apartments is being recorded along with temperature, relative humidity and in some apartments $CO_2$ levels, throughout one summer. Actigraphy records nightly sleep disturbance and every morning an app-based diary captures perceived sleep quality. Questionnaires following spells of hot weather capture changes in sleep pattern, sleep quality, and consequential health and well-being.

**Ethics and dissemination** The study protocol was approved by the Loughborough University ethics committee. The participants will receive both verbal and written information explaining the purpose of the study, what is expected of them, the incentives for participating and the feedback that will be provided. The results will be reported bi-annually to a project advisory board. Presentations will be made at conferences and the methods, intermediary and final results, in academic journals. Informing government bodies, professional organisations, construction industry representatives and housing providers is of particular importance.

## INTRODUCTION

The Homes Heat Health (HHH) project is monitoring a cohort of healthy adults sleeping at home during English summers. The protocol reported here addresses the first two project aims: to understand the relationship

## STRENGTHS AND LIMITATIONS OF THIS STUDY

⇒ For the first time in the UK, the Homes Heat Health project combines measured and self-reported sleep quality with measurements of the bedroom environment and weather conditions to understand the effect of heat on sleep disturbance.
⇒ The research will enable alternative bedroom overheating criteria to be tested and methods of improving summertime sleep in existing buildings to be trialled.
⇒ The participants, who are monitored for one summer, have no pre-existing conditions that would affect sleep quality, span an age range from 25 to 75, and live in London apartments—a building type most at risk of summertime overheat.
⇒ The study collects data in natural settings so relies on participants' engagement and reporting.
⇒ The ambient conditions in people's homes are affected by the external conditions and so the research is vulnerable to the weather during the study periods.

between heat and health; and determine the threshold indoor temperature which causes unacceptable sleep disturbance. The protocol lays the foundation for addressing the third aim: to develop behavioural and technical interventions that can reduce the summer night-time overheating in UK apartment buildings.

### Heat, sleep and UK homes

Good quality sleep is essential for human health and well-being[1] and almost every major disease in the developed world shows a causal link to lack of sleep.[2] Sleep disruption degrades mental health, reduces workplace productivity, increases absenteeism and elevates the risk of accidents.[3] It therefore undermines independent living and increases the burden on health and social

care systems.[4] Both lifestyle and environmental factors lead to sleep disruption. The WHO cites sleep disturbance as one of the most serious consequences of environmental noise and elevated temperatures,[5] with poor air quality also being a factor.[6]

The prevalence of overheating in homes in temperate regions is increasing.[7] 'Higher outdoor or indoor ambient temperatures, expressed either as daily mean or night-time temperature, are negatively associated with sleep quality and quantity worldwide'.[8] In 2017, 4.1 million English households reported that their main bedroom overheated.[9] Those living in apartments and small houses in London and the Southeast of England are disproportionately exposed to higher night-time temperatures. Heat waves are associated with excess mortality and increased morbidity in vulnerable groups including the elderly, young children and those with chronic conditions.[4 5] Heat-related deaths in the UK are expected to more than triple, to 7000 a year by the 2050s.[4] The UK Climate Change Committee has classified overheating in its highest risk to health and well-being category.[10]

Despite the well-known problem, 'there is limited research on thermal comfort in sleeping environment' [sic][11] and in the last 50 years, 'only a few researchers investigated thermal comfort in sleeping people' in natural settings. Likewise, the WHO[5] notes that there are very 'few studies of the direct effect of high indoor temperature on health'. A recent systematic review[12] identified cohort-based studies in Korea,[13] China[14] and Germany[15] involving people sleeping in their own homes. Yet sleep is an area in which the location, dwelling, social and cultural context is influential so, in the absence of a UK study, the credibility of public health advice and the criteria used for assessing overheating in homes is undermined.

### Defining overheating in UK homes

Well-designed homes can provide peak summertime indoor temperatures below those prevailing outside, poorly designed buildings exacerbate the overheating problem; 'protection against outdoor heat is a key characteristic of healthy housing'.[5] While there are UK Building Regulations to reduce overheating risk in new homes,[16] there is no such regulation for existing homes.

Current UK guidelines deem bedrooms to be overheated if the operative temperature (a mix or air and radiant temperature) exceeds 26°C,[17] a criterion based on a 1975 study with just 21 adult participants in one area of England.[18] In the last 45 years, sleeping habits, bedding and nightwear, summer temperatures and dwelling insulation standards have changed substantially. The WHO's recent review[5] flagged the 'need to identify the "maximum acceptable temperature", above which the risk to human health increases'.

### Curbing overheating

The UK Health Security Agency, the WHO and others provide generic advice on staying cool during heat waves. Behavioural change advice includes the temporal use of windows and internal shading devices, especially curtains, but also technical interventions, such as improved shading and ventilation, the use of fans, and suggestions to sleep in a cooler room.[19] Little is said about adaptations to bedwear and bedding to improve sleep quality.

Many factors limit effective adaptation, especially in city apartments at night. Noise, external pollution, security and safety concerns may inhibit bedroom window opening. Older people, who could benefit most from heat-health advice, can find it challenging to effectively operate windows and shading. Those that support people living in their own home, or who manage care settings, often lack the awareness and knowledge to effectively manage others' bedroom environment, bedding and bedwear.

Landlords, local authorities, charities and other housing providers need clear, bespoke advice on cost-effective retrofit solutions, especially those that can be readily integrated into maintenance and repair schedules. Modelling studies have demonstrated the value of heat-protective glazing, external shading, noise-controlled ventilation, the installation of fans and green urban designs in reducing night-time temperatures. However, models are unreliable predictors of overheating in dwellings[20] and there are very few studies examining the practicality, cost and effectiveness of heat adaptation measures.[21 22]

### Contribution to knowledge

While the effects of day-time heat on people is relatively well understood, the effects of night-time heat on sleep quality and health is a nascent area.[23 24] Valuable insights have been, and continue to be obtained from controlled, laboratory trials.[25–29] These have enabled the proposal of a night-time overheating criterion for assessing UK homes.[12] However, very little work has been undertaken in natural settings,[30] the work reported in three studies[13–15] being exceptions, but none of these studies are in the UK. Thus, the HHH project is the first project to measure the relationships between the outdoor environment, the bedroom environment, sleep quality, and the health and well-being of healthy adults in natural UK settings.

Methodological approaches to the assessment of ambient conditions and sleep quality have tended to be indirect and unstandardised.[30] Innovation lies in how we integrate different measurement methods and account for confounding factors, both personal (eg, stress, illness) and environmental (eg, air quality, ambient light), when interpreting bedroom temperature and sleep data (eg, disruption, disturbance). If successful, our research will produce a practical, valid and reliable methodology, measurement methods and criteria for field assessments.

While directly relevant to temperate climates, our study protocol will assist studies of other people, countries, cultural settings, weather conditions and buildings.

## Research hypotheses and aims

The HHH study seeks to understand the relationship between heat and health and determine the threshold indoor temperature which causes unacceptable sleep disruption. Three hypotheses frame our research:

1. A practical measurement protocol, informed by contemporary sleep science, relevant to public health providers and the needs of building designers and regulators, can be devised to quantify temperatures and other factors that cause sleep disturbance in bedrooms.
2. Relevant, operable, behavioural changes and technical interventions can improve the bedroom environment and, thus, sleep quality even during hot weather.
3. Cost effective mitigation measures within building refurbishment cycles can substantially reduce overheating, improve the sleeping environment and reduce energy demand.

The aims of the project are therefore to:

1. Develop a practical, measurable, replicable and clinically meaningful definition of temperature-related sleep disturbance, and to quantify the impact of such disturbance on personal, social and occupational functioning.
2. Produce new, credible overheating criteria based on sleep disturbance for use in identifying overheating in *new* homes and, through measurement, in *existing* homes.
3. Trial co-created socio-technical adaptations and assess their practicality, adoptability and influence on the indoor environment, sleep quality, and health and well-being for people of different ages.

The methodology reported here addresses the first two aims and their associated hypotheses. The key to success is the development of practical, valid and reliable methods of recruitment, sleep monitoring, bedroom surveying and observational field assessments.

## METHODS AND ANALYSIS

The research is being conducted over a period of 33 months with two monitoring campaigns in the summers of 2023 and 2024 (figure 1). A small, short pilot study will be embedded into the beginning of the 2023 monitoring campaign.

The project places a considerable load on the research team during the spring to ensure that the recruitment phase does not encroach into the summer months. To manage this workload, additional field workers will be employed.

## Participants

The target cohort comprises healthy adults, living in apartment buildings, a dwelling type which is particularly prone to high summertime bedroom temperatures. To aid logistics, participants will be recruited from a limited set of buildings clustered in a constrained area of London.

Eligible participants must meet all the inclusion criteria:

► Be capable of providing informed consent freely and voluntarily.
► Reside in an apartment within the designated recruitment area.
► Be between 25 and 75 years of age.
► Have the intention to remain in their current dwelling for at least 2 years.
► Have access to a personal smartphone equipped with internet connectivity.
► Be willing to download and use the study's proprietary sleep diary app.

Participants were excluded if they met any one of the following criteria:

► Presence of air conditioning in the apartment.
► Past or current diagnosis of psychiatric or sleep disorders.
► Use of prescribed medication intended to induce sleep.
► Presence of symptoms indicative of low mood.
► Poor mobility.
► Pregnant, breastfeeding or caring for children under the age of 2 years.
► Undertaking night shift work.
► Habitually engaging in international travel.

The exclusions criteria help to mitigate potential confounding factors and ensure the safety and well-being of participants, focusing the study on a population whose sleep patterns are less likely to be influenced by external variables or pre-existing medical conditions.

To aid recruitment, contacts in local authorities, housing developments, housing associations and elsewhere will

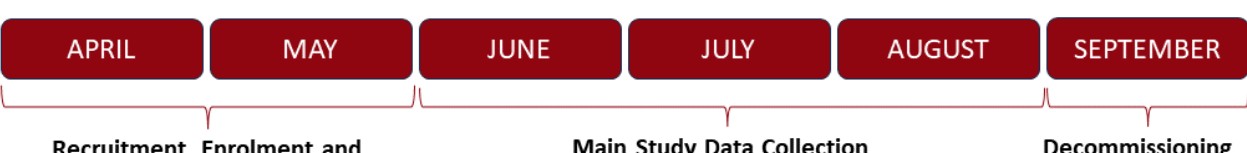

**Figure 1** Study timeline.

provide a route to potential participants. Flyers, posters and a newsletter will be distributed with a QR link to an online screening questionnaire. A social media campaign via Meta will run alongside the 'paper' recruitment using data analytics to reach individuals aged 25–75 living in East London. A website and video will provide more information and an incentive of £150 is offered to the selected participants.

## Sample size

The required number of participants is estimated on the basis of sleep efficiency data reported in earlier studies. Measured as ((total sleep time/time in bed)×100), sleep efficiency has been shown to provide a sensitive marker of temperature-related change in sleep structure resulting from both normal seasonal variations[31] and humid heat exposure.[32] Instrumental measurements of sleep efficiency have also been shown to correlate significantly with subjectively reported sleep quality.[33] Setting alpha at 0.01 (appropriate for a primary outcome) and beta at 80%, and assuming a correlation between repeated baseline/heatwave sleep efficiency measurements of 0.05, and an effect size of at least 0.2 (both informed by earlier research[32 33]), the study requires a minimum sample of 97 participants. To accommodate attrition over two consecutive summers, a target baseline sample of 150 participants is set, covering the summers of 2023 and 2024.

## Participant screening and baseline characterisation
### Screening

The screening questionnaire (table 1) will be the first point of contact between participants and the study team. Like all other questionnaires, it is hosted online via the Qualtrics survey management platform.[34]

First, the questionnaire elicits voluntary informed consent, explains the purpose and duration of the study, the required monitoring and diary keeping, the feedback and incentives that will be provided, and ethical and data anonymisation and protection matters. Second, screening questions elicit the information necessary to determine suitability for inclusion in the study (table 1). Finally, the questionnaire notes that researchers will visit eligible participants at the start of the monitoring period to answer questions, take measurements of their bedroom and install the monitoring equipment.

On completion of the questionnaire, a holding message, and confirmation that the responses have been received, will be sent to the prospective participant. The screening data are then manually reviewed by a researcher and participants who meet the inclusion criteria are formally enrolled to the study, imported to a research log and allocated a unique participant ID.

### Baseline

Following acceptance on the study, a baseline questionnaire will be completed which seeks to characterise each participant and their household (table 1). This enables

**Table 1** Overview of screening and baseline questionnaires

| Topic | Data description |
|---|---|
| **Screening questionnaire** | |
| Purpose, protocol, ethics | Informed consent of participant |
| Participant profile | Name and contact details<br>Age, sex at birth and gender |
| Apartment requirements | Intended occupancy over the next 2 years<br>Presence or absence of air-conditioning |
| IT requirements | Available internet access<br>Availability of smart phone |
| Lifestyle stability | Presence of young children or other caring responsibilities, pregnancy status<br>Typical work patterns and international travel |
| Sleep and health | Weight, height, neck circumference<br>Currently diagnosed with/treated for sleep disorder<br>Obstructive sleep apnoea risk[46 47]<br>Symptoms of Restless Legs Syndrome[48] |
| Mental and physical health | Frequency of negative feelings (PHQ-9)[49]<br>Reported mobility/physical health problems and prescribed medication |
| **Baseline questionnaire** | |
| Participant demographics and long-term health | Education, employment status, ethnicity, smoking, mental health or illness, ability to undertake daily tasks |
| Household characteristics | Number in household and ages of household members<br>Length of time living in the apartment |
| Sleeping behaviour | Sleep posture and bedwear, bedding and body coverage<br>Windows, curtains and ventilation |
| Health Utilisation Index (HUI) | Frequency of use of GPs, A&E units and pharmacies |
| Pittsburgh Sleep Quality Index (PSQI)[50] | Timing of sleep period<br>Reasons and frequency of sleep difficulties<br>Impact on daytime functioning |
| Morningness or Eveningness Questionnaire (MEQ)[51] | Preferred time of waking and sleeping<br>Self-assessed morning or evening classification |
| Insomnia (from DSM-5)[41] | Difficulty of sleeping<br>Impact on daily functioning<br>Use of sleep-assisting medication |
| Ford Insomnia Response to Stress Test (FIRST)[52] | Likelihood of different stressors affecting sleep |
| Functional Outcomes of Sleep Questionnaire (FOSQ)[53] | Difficulty of undertaking different activities due to sleepiness |
| Quality of life (EQ5D)[54] | Mobility, self-care, routine activities, pain/discomfort, mood and general health |
| Warwick-Edinburgh Mental Well-being Scales (WMWBS)[55] | Feelings and thoughts and frequency of these |

the sample to be compared with the UK population as a whole and the cohorts used in others' trials.

The questionnaire also characterises participants' general health and well-being, health-related quality of life, mood state, sleep-related preferences and practices, and individual differences relevant to sleep quantity and quality. The questionnaire data will also enable individuals' sleep to be characterised and provide the benchmark against which to measure subsequent changes (table 1).

### Bedroom and apartment survey and monitoring

Visits will be made to each participant to survey the apartment and bedroom, install the bedroom environment monitoring equipment and guide the participants in the use of the actiwatch and the sleep app with its nightly sleep and weekly health questions. The visit is an opportunity to answer participants' questions and explain the study in a more personal setting. Participants will book their appointment at a time convenient to them via Calendly.[35] The researchers will undertake visits in pairs to provide chaperoning and assistance with the monitoring and surveying work.

#### Bedroom and apartment survey

The survey is undertaken to understand the scope for behavioural changes that could improve sleep quality in hot weather and the opportunities for making physical adjustments to reduce bedroom temperatures. The survey entails physical measurements, making observations and taking photographs (table 2).

To ensure all the necessary data are captured, the surveyors will use a standard form accessed through a digital note pad. All the surveyors will be trained and a trial measurement exercise undertaken prior to their deployment in the field. The aim is to achieve consistent and accurate coding by all the surveyors.

The measurements and observations will be transferred into a standard Excel workbook and then independently checked to ensure they are internally consistent and match the photographic evidence. Erroneous data will be corrected but, if necessary, the apartments will be re-surveyed.

#### Bedroom environment monitoring

Combined temperature and relative humidity sensors, HOBO MX1101,[36] will be placed in each bedroom, but in approximately one-third of the bedrooms HOBO MX1102A temperature, relative humidity and carbon dioxide ($CO_2$) sensors[36] will be used (sensors' accuracy ±0.21°C, ±2% rh, ±50 ppm $CO_2$). Night-time $CO_2$ levels provide an indication of air-quality and how well the bedroom is ventilated.

The sensors will be set to record continuously at half hourly intervals, and placed by the researchers on a convenient surface out of direct sunlight, away from other sources of heat, and no more than 1.8 m above floor level (the zone occupied by people). Participants

will be asked where the sensor could be placed so as not to inconvenience them; they will be asked not to move it.

### Measured and subjective sleep monitoring

Sleep will be continuously monitored throughout the study period using wrist actigraphy and a daily sleep diary.

**Table 2** Overview of bedroom and apartment survey

| Data category | Data description |
| --- | --- |
| General | Surveyors' names<br>Participant ID |
| Apartment description | Type: Single level, duplex, studio-apartment<br>Number of rooms<br>Type of each room: bedroom, living room, bathroom, etc<br>Heating system type<br>Sources of internal heat, for example, HIU,* hot water tanks†<br>Natural ventilation: single sided, cross-vented‡<br>Mechanical ventilation: extract only, supply and extract,§ MVHR¶ |
| Bedroom description | Floor plan, dimensions, bed location, heat emitter** locations, ceiling height<br>Volume and location of large wardrobes, cupboards, etc<br>Natural ventilation: single sided, cross-vented‡<br>Mechanical ventilation inlets and outlets<br>Presence of fans, dehumidifies, portable air-conditioning |
| Bedroom windows | Orientation of each window (degrees from north)<br>Number of windows and the dimensions of each<br>Dimensions and type of each glass area (single, double, etc)<br>Number of opening windows, type of opening and opening area |
| Bedroom shading | Presence and dimensions of external shading: overhangs, side-fins<br>Internal shading for example, curtains (thick or thin), roller blinds, louvre blinds, etc |
| Photographs | External elevation of building showing apartment location<br>Views from bedroom window(s)<br>Window and curtain details<br>Hot water tanks and other internal heat sources<br>Fans, dehumidifier, air-conditioning if present<br>Mechanical ventilation inlets and outlets |

*HIU transfers heat from the building-wide heat loop into the apartment.
†Hot water tanks commonly store hot water for washing and may also be used in conjunction with the HIU as part of the space heating system.
‡Bedrooms and apartments that have single sided ventilation have windows and other openings facing in only one direction. Cross-ventilated apartments and bedrooms have openings on opposite sides of the building/bedroom. Cross-ventilation provides more effective natural night-time cooling of indoor spaces.
§Mechanical ventilation may be by extracting air (typically from kitchen and bathroom areas) or by extracting air and supplying fresh air to living spaces.
¶MVHR units balance extract and supply air and transfer heat from the exhaust air to the supply air in winter.
**Either an electric heater or a radiator in a water-based system.
HIU, heat interface unit; MVHR, mechanical ventilation heat recovery.

**Table 3** Sleep outcome metrics measured by actigraphy and reported via the daily sleep diary

| Sleep outcome | Data description | Measured | Reported |
|---|---|:---:|:---:|
| Sleep location | Did participant sleep in own bed? | | ● |
| Time in bed (TIB) | Total time between getting into bed and getting up | ● | ● |
| Sleep onset latency (SOL) | Time elapsed between 'lights out' and the onset of sleep | ● | ● |
| Total sleep time (TST) | Total time asleep in bed | ● | ● |
| Sleep period time (SPT) | Time elapsed from first onset of sleep to final awakening | ● | |
| Intervening awake periods | Number of episodes of waking up from sleep | ● | ● |
| Intervening awake duration | The percentage of time in bed spent awake | ● | ● |
| Sleep efficiency (SE) | Percentage of time in bed in recorded sleep, (TST/TIB)×100 (%) | ● | ● |
| Sleep fragmentation index* | Time mobile and immobile, the index provides a measure of restlessness during sleep | ● | |
| Average light intensity | Intensity of bedroom light during the night (lux) | ● | |
| Central phase measure | Midpoint between 'fell asleep' and 'wake up', expressed as the number of minutes past midnight | ● | |
| Sleep quality rating | 1–5 Likert scale: 1=very poor; 5=very good | | ● |
| Rested/refreshed on awakening | 1–5 Likert scale: 1=not at all rested; 5=very rested | | ● |
| Thermal comfort in bedroom | 1–5 Likert scale rating: 1=extremely comfortable; 5=not at all comfortable | | ● |
| Sleep disturbance attributed to temperature | 1–5 Likert scale rating: 1=extremely disturbed; 5=not at all disturbed | | ● |
| Nap duration | Total time spent napping in the previous day | | ● |

*The sleep fragmentation index is calculated as (percentage of the TST categorised as 'mobile' in epoch-by-epoch analyses)+(the number of immobile bouts ≤1 min expressed as a percentage of the total number of immobile bouts in the TST).

## Actigraphy

Actigraphic recordings will be provided by the wrist-worn Motionwatch 8 actiwatch[37] which is a research-grade triaxial piezoelectric accelerometer that detects movement in all planes. Since activity-rest patterns provide a close analogue of wake-sleep patterns, accelerometer signals can be processed to yield measures of sleep onset, continuity and duration[38] (table 3). The Motionwatch also incorporates a light sensor which records daily photopic white light illuminance.

Each Motionwatch will be set to record in 1 min epochs and given to participants during the apartment survey visit. Advice on its use will be given both verbally and in writing. Participants will be asked to wear the watch on their non-dominant wrist for the duration of the study (except when washing) and to log any deviations from the agreed protocol (eg, holiday times, non-wear times). Actigraphy data will be processed using the proprietary software MotionWare (V.1.3.35).[39]

## Daily sleep diary

The daily sleep diary is a phone app (branded 'The City Sleep Diary'), which has been developed for this study and is a pivotal tool for augmenting the actigraphy data and capturing participants' sleep experience. The app's content is based on the widely validated Consensus Sleep Diary[40] with additional questions on sleep-related thermal comfort and a check that participant had slept in their own bedroom the previous night. The diary reports enable key sleep outcome metrics to be derived (table 3).

Participants will be asked to download the app from the Google Play or App Store prior to the baseline home visit using a link provided to them. During the apartment survey visit, the team will provide a demonstration of the app, emphasising the importance of completing the diary each morning. App features include: a progress bar displaying completion levels; two background light-intensity modes; an FAQs section; and a 'Contact Us' feature enabling direct email communication with the research team.

## Weekly health tracker

To monitor each participant's health status, the visual analogue health self-rating from the EQ5D-5L[41] is added to the sleep diary questions once every week. Participants will move an analogue 'slider' to a point on the scale corresponding to their weekly health rating (0: the worst imaginable health state, 100: the best imaginable health state). They are asked to complete the tracker once a week but given flexibility to choose the preferred day.

## Mid-study visit

In the middle of the main study period (see figure 1) willing participants will be visited at a time identified

**Table 4** Overview of hot spell and end-of-study questionnaires

| Topic | Data description | Hot spell | End-of-study |
|---|---|:---:|:---:|
| Sleeping behaviour | Sleep posture<br>Bedwear, bedding and body coverage<br>Windows, curtains and ventilation | ● | ● |
| Health Utilisation Index (HUI) | Frequency of GP, A&E and pharmacy usage | ● | ● |
| Functional Outcomes of Sleep Questionnaire (FOSQ) | Difficulty of undertaking different activities due to sleepiness | ● | ● |
| Quality of life (EQ5D) | Mobility, self-care, routine activities, pain/discomfort, mood and general health | ● | ● |
| Pittsburgh Sleep Quality Index (PSQI) | Timing of sleep period<br>Reasons and frequency of sleep difficulties<br>Impact on daytime functioning | | ● |
| Warwick-Edinburgh Mental Well-being Scales (WMWBS) | Feelings and thoughts and frequency of these | | ● |
| Participant experience | Participant satisfaction, likelihood of recommending the study to others, participation in future, related research | | ● |

by them in Calendly. The primary motivation is to fit a new actiwatch battery and download the actiwatch data; the watches have limited capacity. During the visit, the temperature data and, if present, $CO_2$ data will also be downloaded. The visit also enables the bedroom and apartment survey data to be checked and offers participants another opportunity to engage with the research team.

### Hot spell questionnaire

An online questionnaire will be delivered after periods of hot weather to identify the effects of heat on sleep (table 4). There is no formal definition of a hot spell, so the questionnaire will be delivered immediately after or during hot weather periods of different duration and severity. Warnings are provided by the heat-health alerts issued by the UK Health Security Agency and Met Office[42] which classify imminent hot weather on a green, yellow, amber or red scale based on the likelihood of occurrence and the potential impacts.[43] Amber or red classifications are of particular interest in this study.

The hot spell questionnaire will take a snapshot of sleeping behaviour during the previous hot nights via questions compiled specifically for this study (table 4). Health and mental functioning will be captured using a reduced set of the questions in the baseline questionnaire (table 4). So as not to overburden participants, the questionnaire will be delivered no more than four times per summer.

### End-of-study questionnaire

An end-of-study online questionnaire will be administered which, like the hot weather questionnaire, will contain a subset of the questions in the baseline questionnaire (table 4).

End-of-study visits will be arranged throughout September at a time that participants identify in Calendly and will last 15–20 min. The purpose is to remove the

sensors, debrief the participants and formally end their involvement in the active phase of the study. Data from the temperature monitoring equipment and actiwatch will be downloaded during the visit to reduce the risk of data loss during transit.

### Analysis plan

The raw data from the bedroom survey and questionnaires will be digitised and, along with the temporal data from the actiwatches and environment monitors, will be stored on secure servers at Loughborough University. Descriptive statistics profiling the characteristics of the participants and apartments will be presented as frequencies, proportions and means with appropriate indices of dispersion.

The temporal data will be cleaned, that is, gaps and erroneous values filled, corrected and/or flagged, and data items labelled. For each night, the metrics that describe sleep quality and quantity (see table 3) and the bedroom environment, for example, mean night-time temperature, relative humidity and $CO_2$ level, will be calculated. The nightly metrics will be organised in Excel files for subsequent analysis, for example, using Matlab, R, SPSS or other analysis and statistical packages.

To assess the impact of hot weather spells on sleep and health-related outcomes (project aim 1), the nightly sleep and environment metrics data will be averaged across a 7-night period of cool summer weather. These metrics provide the baseline against which to compare the corresponding metrics for each hot spell. The baseline and hot spell metrics can then be compared in univariate general linear models for repeated measures. Model covariates will be fitted as appropriate.

To characterise the way in which participants' sleep quality and quantity varies with temperature, the mean night-time bedroom and outdoor temperatures will be compared with the nightly sleep metrics for the whole

of each summer monitoring period. The derived regression and time-series models will describe the percentage of participants experiencing sleep disturbance at any given temperature. The threshold bedroom and outdoor temperatures at which a chosen percentage of people experience unacceptable sleep disturbance, that is, the overheating thresholds, can then be derived (project aim 2).

Further analysis will try to establish the bedroom and participant characteristics that have a significant impact on sleep disturbance, but the extent of such analysis may be constrained by the limited cohort size.

Anonymised data that can be made publicly available without breaching participant confidentiality will be made available through the Loughborough University Institutional Repository.

### Participant and public involvement

Project partners and our study participants will help to shape and steer our research. The project partners are drawn from central government departments, charities, local authorities and housing associations. The study tools and methods have been designed and developed in consultation with our partners and piloted with London residents. Feedback has informed the main study design, which was given a public-facing name: 'Sleep in the City'.

### Ethics and dissemination

The study protocol has been approved by the Loughborough University ethics committee (ref. 2023-14123-13637). The participants will receive both verbal and written information explaining the purpose of the study, what is expected, and the incentives on offer. Participants will be assigned an ID number and all collected data will be stored securely on password-protected servers at the University. This will be held for 5 years after the study ends in compliance with the ethical and legal practice required by the UK Data Protection Act (1998).[44] Access to the file linking participants' IDs to their personal details will be restricted. Where participant consent is given, anonymised data will be deposited in the University repository so that it can be made publicly available for future research.

The study progress will be reported twice a year to the project partners, who also offer a direct route to impact. The results will be reported in international peer-reviewed journals concerned with climate change, sleep, health, and building design, and at conferences and seminars in these fields. Ad-hoc presentation will be given in response to specific requests and press releases issued as appropriate. Anonymity of the participants and the buildings in which they sleep will be maintained at all times.

## DISCUSSION

To our knowledge, this is the first UK study to measure the relationships between the outdoor environment, the bedroom environment, sleep quality, and the consequential health and well-being of adults sleeping in their own home. The project is complex, requiring the deployment of a range of convenient and unobtrusive methods of measurement and monitoring drawn from different fields of endeavour: engineering, sleep science, human behaviour, and health and well-being. A multi-disciplinary team has been brought together to manage this complexity.

The research has the potential to shape public health advice, influence the design of apartment buildings, and identify appropriate and effective interventions to improve summertime sleep.

The research protocol and methods could be readily adopted in other climatic, cultural and construction contexts where there are concerns about the effects of heat on human health.

### Limitations

Given the aim of studying people in a natural setting, the project funding and duration, and yet a wish to isolate the effect of heat on sleep from the many other influential factors, the cohort is restricted to healthy, IT-literate individuals who have robust sleeping patterns and live in dwellings that are likely to overheat in the summer, that is, London apartments. These restrictions could skew the age group of the sample, their sleeping habits and the way they manage their bedroom at night, however, we anticipate this skew will align with the demographic spread[45] and lifestyle of people living in London apartments.

The project will provide evidence that relates directly to 'real life', so participants are at liberty to change their behaviour and bedroom environment during the study period. The daily sleep app will identify if participants have not slept in their own bed, and the heat wave questionnaire will ascertain if people have changed their behaviour or are feeling unwell. The analysis will therefore be able to account for these factors. The study will identify if participants feel unwell on a weekly basis, but short periods of illness, stress or other naturally occurring factors that affect nightly sleep may go undetected.

Environmental factors such as noise, light levels and perhaps pollution also affect sleep. While the general noise and pollution environment around each apartment will be determined from publicly available data sources, the nightly variations in these cannot be captured cost effectively. The ambient light levels are measured using the actiwatch but this is an unreliable measure of the ambient bedroom light level.

Researching the effects of hot spells in the UK is particularly problematic. A cooler summer with no amber or red heat-health warnings will preclude a focused examination of the influence of hot spells on sleep quality. Conclusions about the effects of temperature could still be possible but be limited to lower temperatures where few, if any, effect on health and well-being are observed.

The research tools have widespread applicability for quantifying the effect of heat on sleep, however, care must be taken if questionnaires are converted to other

languages; adjectival descriptions can have subtly different meanings in other linguistic and cultural contexts.

While human thermal physiology may be very similar across the globe,[8 11 12] social and personal contexts are not—and these influence our sleeping habits (bedtime, bedwear, bedding, bedroom thermal management, etc) and hence the temperatures, and temperature changes that may result in sleep disruption. While the analysis methods will attempt to disentangle the various factors affecting sleep disturbance the size of the cohort limits the insights possible.

The conclusions of this project will be directly relevant to city-dwellings in other temperate regions who live in non-air-conditioned dwellings. Extrapolation to other contexts and to hotter or colder regions must, however, be undertaken with care. There is a need therefore for further studies in different social and geographical contexts.

**Acknowledgements** The authors thank the project partners for their advice and support.

**Contributors** The study was conceived by and is led by KL. KM contributed sleep-science knowledge to the funding bid. AB and KL, supported by VR, devised the apartment survey and monitoring methods. KM and IH, with support from MR, devised the sleep monitoring methods, sleep diary app and weekly questionnaires. VH, JB and ZZ devised the screening, baseline, hot spell and end-of-study questionnaires. The first draft of this paper was created by KL, which all authors critically revised and approved prior to submission.

**Funding** The study is funded by the UK Engineering and Physical Sciences Research Council and National Institute for Health Research (Grant EP/W031736/1) as part of the UK Research and Innovation initiative 'Transforming care and health at home and enabling independence'.

**Competing interests** None declared.

**Patient and public involvement** Patients and/or the public were involved in the design, or conduct, or reporting, or dissemination plans of this research. Refer to the Methods section for further details.

**Patient consent for publication** Not applicable.

**Provenance and peer review** Not commissioned; externally peer reviewed.

**ORCID iDs**
Kevin Lomas http://orcid.org/0000-0001-5792-0762
Kevin Morgan http://orcid.org/0000-0001-6755-2502
Victoria Haines http://orcid.org/0000-0002-2722-9951
Iuliana Hartescu http://orcid.org/0000-0001-5125-9096
Arash Beizaee http://orcid.org/0000-0003-1480-3480
Jo Barnes http://orcid.org/0000-0002-3291-8006
Zoe Zambelli http://orcid.org/0000-0001-6604-213X
Monisha Ravikumar http://orcid.org/0000-0002-7126-1879
Vincenzo Rossi http://orcid.org/0000-0002-6653-9071

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
