## [Reviewer comments · BMJ Open]

ARTICLE DETAILS

TITLE (PROVISIONAL)	Homes Heat Health protocol: an observational cohort study measuring the effect of summer temperatures on sleep quality
AUTHORS	Lomas, Kevin; Morgan, Kevin; Haines, Victoria; Hartescu, Iuliana; Beizae, Arash; Barnes, Jo; Zambelli, Zoe; Ravi, Monisha; Rossi, Vincenzo

VERSION 1 – REVIEW

REVIEWER	Hoe, Victor University of Malaya, Kuala Lumpur
REVIEW RETURNED	19-Apr-2024

GENERAL COMMENTS	The Homes Heat Health protocol is an observational cohort study aiming to measure the impact of summer temperatures on sleep quality. The study focuses on understanding the relationship between heat and health, determining the indoor temperature threshold causing sleep disruption, and developing interventions to reduce overheating in apartment buildings. The research involves monitoring healthy adults in London apartments during summers, collecting data on sleep patterns, bedroom environment, and weather conditions. The study protocol includes participant screening, baseline questionnaires, actigraphy for sleep disturbance, app-based diaries for sleep quality, and weekly health tracking. The protocol emphasizes the importance of quality sleep for overall health and well-being, highlighting the negative effects of sleep disruption on mental health, productivity, and overall quality of life. The study aims to provide valuable insights into the impact of temperature on sleep quality and health, particularly in the context of increasing overheating in homes, especially in urban areas like London. Overall, the protocol has provided an in-depth description of the study. However, there are two areas that can be improve: 1) Proposed or planned analysis2) A more in-depth discussion on the limitation and how the limit
--

REVIEWER	Lan, Li Shanghai Jiao Tong University
REVIEW RETURNED	06-May-2024

GENERAL COMMENTS	The environmental quality of real bedroom and thus sleep is affected by multiple factors, including temperature, humidity, noise, and IAQ etc. Thus in this cross-sectional study, it is difficult to find the threshold indoor temperature which causes unacceptable
---

	sleep disruption. The authors are suggested to analyze the data with appropriate statistical method.
--	--

VERSION 1 – AUTHOR RESPONSE

Reviewer 1: Proposed or planned analysis	A section describing the planned analysis has been added.
Reviewer 1: A more in-depth discussion on the limitation and how the limit	Some limitations of the study were included in the original discussion section. These have now been repositioned and reorganised, along with some other limitations, in a new section “Limitations”.
Reviewer 2: The authors are suggested to analyze the data with appropriate statistical method.	A section describing the planned analysis has been added.
Lead author’s comments.	Additional small edits have been made as a consequence of the reviewers’ and editor’s comments and, elsewhere, to improve the flow and grammar of the paper. The requested additions have increased the word count to 4,413 words.